# Minimally Invasive Tissue Sampling via Post Mortem Ultrasound: A Feasible Tool (Not Only) in Infectious Diseases—A Case Report

**DOI:** 10.3390/diagnostics13162643

**Published:** 2023-08-10

**Authors:** Akhator Terence Azeke, Julia Schädler, Benjamin Ondruschka, Stefan Steurer, Dustin Möbius, Antonia Fitzek

**Affiliations:** 1Department of Anatomic Pathology, Irrua Specialist Teaching Hospital, KM 87 Benin Auchi Rd, Irrua 310115, Nigeria; 2Institute of Legal Medicine, University Medical Center Hamburg-Eppendorf, Butenfeld 34, D-22529 Hamburg, Germany; 3Institute of Pathology, University Medical Center Hamburg-Eppendorf, Martinistraße 52, D-20246 Hamburg, Germany

**Keywords:** forensic pathology, minimal invasive tissue sampling (MITS), ultrasound, post mortem, conventional autopsy, COVID-19, histopathology

## Abstract

In the past years the number of hospital autopsies have declined steadily, becoming almost excluded from medical training. Medicolegal (forensic) autopsies account for almost all autopsies, whereas hospital autopsies are becoming increasingly rare. Minimally invasive tissue sampling (MITS) using post mortem ultrasound offers the opportunity to increase the number of post mortem examinations in a clinical and even forensic context. MITS is a needle-based post mortem procedure that uses (radiological) imaging techniques to examine major organs of the body, acquire tissue samples and aspirate fluid from the body cavities or hollow organs. In this study, MITS was used to determine the presence of other co-existing diseases in a deceased infected 97-year-old woman with severe acute respiratory syndrome coronavirus 2. The examination of her body was carried out using ultrasound as an imaging tool and to gather ultrasound-guided biopsies as conventional autopsy was rejected by the next of kin. Ultrasound and histology identified an intravesical mass leading to an obstruction of the urinary outlet resulting in bilateral hydronephrosis and purulent pyelonephritis, which was unknown during her lifetime. Histopathological examination revealed the tumor mass to be a squamous cell carcinoma. This study has shown that MITS can be used to determine the cause of death and the presence of concomitant diseases in the infectious deceased.

## 1. Introduction

Minimally invasive tissue sampling (MITS) is a needle-based technique that combines imaging efforts such as ultrasound, magnetic resonant imaging, and computed tomography, with a biopsy approach aimed at collecting samples of the main organs and body fluids post mortem without opening the body as done in conventional autopsy [1,2].

Information acquired from post mortem examinations of patients infected with high-risk pathogens is an important part in directing a proper response to pandemics, especially in their early phases [3,4]. Autopsy allows for a rigorous structured recording of comorbidities for risk assessment as well as the identification of important pathophysiological and molecular mechanisms in organs that could be important targets for future therapeutic interventions [3]. However, the 2019 coronavirus disease pandemic has shown that, due to the classification of severe acute respiratory syndrome coronavirus 2 (SARS-CoV-2) as a risk group 3 (RG-3) pathogen by the European Commission and several other organizations, there were initial concerns about performing autopsies due to biosafety risks [5]. Therefore, the acquisition of a biosafety level 3 or 4 autopsy room is advisable in developing countries, where many high-risk diseases occur. However, the establishment of such facilities is far beyond the means of low- and middle-income countries [6].

Very recently, Autsch et al. [7] summarized all published conventional autopsy results of SARS-CoV-2-associated deaths above *n* > 10, which were published mostly by European countries, while the highest count worldwide (*n* = 283) was presented by the author’s group. However, there were also different attempts in reducing the (possible) transmission of virus-containing aerosols by using radiological, MITS-based or laboratory approaches alongside conventional autopsies. While SARS-CoV-2 results in typical, but no pathognomonic changes in chest imaging both pre and post mortem [8,9] it was used in large-sized post mortem evaluations as one additional source of scientific information regarding the causes of death [4]. Detailed virologic assessment from nasopharyngeal swabs, body surfaces, corneal epithelium, and others further allowed detailed insights in the post mortem management of corpses infected by SARS-CoV-2 (and potentially beyond), the viability of the virus after patient’s death, and questions regarding post mortem tissue donation [10,11,12,13,14]. However, laboratory methods only are not useful in answering questions concerning causes and manner of death in detail.

Therefore, while conventional autopsy is the gold standard for studying how disease affects different organs and systems [15,16], MITS has been repeatedly validated as an alternative with a concordance rate to conventional autopsy results of up to 90% in Mozambique and up to 97% in the Netherlands [1,17,18].

Although conventional autopsy offers more information, especially taking into account macroscopic and haptic aspects like color and tissue textures, the number of autopsies has decreased worldwide over the years, e.g., due to cultural and religious practices and psychosocial considerations related to death, as well as the belief that autopsies have lost their value due to advances in pre mortem diagnostics [19,20].

In this respect, MITS via post mortem ultrasound is a suitable alternative to conventional autopsy, as it is not only accepted by the public, but also a desirable technique for pathologists to examine patients with highly infectious diseases [2]. This case report highlights the usefulness of MITS in determining the cause of death in a patient infected with SARS-CoV-2 whose next of kin declined a conventional autopsy but agreed to minimally invasive procedures post mortem.

### Case Report—Clinical History

A 97-year-old woman presented in the emergency department with macrohematuria occurring for the first time. Her known comorbidities were arterial hypertension, coronary artery disease, and dementia. Her initial laboratory tests showed a very low glomerular filtration rate (3 mL/min), highly elevated creatinine (10.7 mg/dL), and markedly elevated c-reactive protein (123.8 mg/L). The leucocyte count was 19.8/nL and potassium was 6.54 mmol/L. Additionally, she also presented with metabolic acidosis with a pH of 7.25. Laboratory results were compatible with the diagnosis of an acute renal failure. Further diagnostics during lifetime, e.g., an ultrasound or CT, were not performed.

During admission, she was tested positive for SARS-CoV-2 via qPCR (throat swab), though she had no suggestive symptoms. A urinary catheter was put in place to ensure continuous bladder drainage and to prevent clot formation. Because of her steady deterioration in general condition she was placed in palliative care after receiving consent from her relatives and legal guardian for this procedure. At that moment, a decision was made for palliative supportive therapy without any further diagnostics and interventions. As a result, the cause of macrohematuria was not further clarified pre mortem. A malignancy was not known during her lifetime. Her condition worsened rapidly and she died two days after admission. A MITS was requested by her relatives to confirm whether she died from or with SARS-CoV-2 but a conventional autopsy was rejected.

## 2. Materials and Methods

After admittance to the Institute of Legal Medicine (ILM), the body was screened for viral SARS-CoV-2 RNA using a throat swab followed by immediate RT-qPCR at the Institute of Microbiology, Virology and Hygiene, UKE, as previously described [21]. Even though there was no order from the prosecutor’s office to open the body after the manner of death was classified as ‘unclear’, the relatives questioned both the pre-existing conditions and the cause of death. About eight hours prior to the post mortem (ultrasonic) examination, her body was removed from the cooling chamber to warm up the soft tissue. During the procedure starting 72 h after death, single-use gowns and gloves, protective hoods, and FFP2/3 masks for individual protection were used by the investigators. The examination of the body was carried out in a supine position. First, an external inspection of the body was performed. Following that, a LOGIQe ultrasound system (LOGIQe 5417728-100, GE Medical Systems Ultrasound and Primary, USA) was used for post mortem ultrasonic examination, according to a standardized protocol developed at the ILM. The ultrasound protocol for post mortem workup was established by our study group within the DEFEAT PANDEMIcs project and further proofed during the NATON period, both funded by the German BMBF (01KX2021 and 01KX2121) for sampling and investigating organs in a standardized manner. It has become an integral part of our institutional standard. The protocol is available from the corresponding authors upon reasonable request.

Subsequently, a post mortem ultrasound-guided needle biopsy of the organs—upper and lower lobes of the right and left lung, septum of the heart, right lobe of the liver, upper pole of the kidneys, central part of the spleen and urinary bladder—using sealed 14G needles (SOMATEX, Berlin, Germany), as well as aspiration of body fluids (e.g., urine) were carried out by depicting the aimed organ and taking needle biopsies under visual control. For each site, at least two samples for histological processing were collected to avoid error sampling. Tissue samples were promptly fixed in buffered 4% formaldehyde and consecutively processed, sectioned, and stained histologically using only hematoxylin-eosin staining (HE). The post mortem ultrasonic examinations and image interpretation were carried out by three investigators as per full consensus (JS, AF, DM—residents in the ILM, including several years of internal medicine education). The duration of the whole procedure was about 60 min.

Ultrasonography of the lungs was performed. Post mortem ultrasound findings were principally consistent with general signs of pneumonia as seen in living SARS-CoV-2-infected patients, with mainly consolidations, enhanced B-lines (multifocal, confluent), an aerobronchogram, and thickened or fragmented pleural lines visible [22].

Histological criteria of COVID-19 were defined as diffuse alveolar damage (DAD), especially in the form of hyaline membranes, activated pneumocytes, squamous metaplasia or organizing pneumonia, as described before.

## 3. Results

### 3.1. Post Mortem Ultrasound Findings

Post mortem ultrasound examinations, according to the standardized protocol developed at the ILM (see material and methods), were performed. The lung ultrasound examination showed small to moderate pleural effusion on the right side while no effusion was detected on the left. Both right and left pleural lines were thickened and fragmented. Posteriorly, B-lines could be detected on both sides (Figure 1a).

Both kidneys appeared considerably smaller than normal with reduced cortices. The right kidney showed an intensively dilated pelvis and the left kidney a massive dilatation of the pelvis (Figure 1b) indicating hydronephrosis. Both ureters were notably dilated. The fluid in both dilated pelvises was aspirated under ultrasound guidance. The fluid in the right kidney pelvis was white-yellow, murky, and purulent, the fluid in the left kidney pelvis was straw colored. Inside the right ureter a visible tumorous mass could be depicted (Figure 1c).

The urinary bladder was distended and filled with a large amount of urine. A second irregular-shaped mass at the posterior and inferior bladder margin could be detected and was a suspected malignancy, forming tamponade-like aspects with the irregular thickened wall of the bladder.

Punch biopsies were taken at each organ as described above.

### 3.2. Histology Findings

Sections of the lungs showed a marked edema with fluid within the alveoli, blood congestion, fibrosis, and dystelectasis. There were no relevant inflammatory changes such as DAD. Typical changes due to pneumonia were not detected.

The heart showed myocardial scars and fibrosis as signs of arterial hypertension and chronic ischemia whereas the liver showed a small droplet fatty degeneration (<10%). Both kidneys presented a chronic interstitial nephritis with severe glomerulosclerosis and hyaline cylinders in tubules (Figure 2a) [23]. The malignant tissue of the urinary bladder consisted of admixed high-grade carcinomatous and sarcomatous elements, compatible with a squamous cell carcinoma (Figure 2b,c).

In summary, in the post mortem ultrasound only fragmented, thickened pleural lines were visible as a hint of pulmonary change. Taken together with the histological examination there are no signs of DAD or pneumonia, ruling out the relevant influence of the SARS-CoV-2 infection on the cause of death. B-Lines were sono-morphologically detected posteriorly after death. The leading diagnostic finding was a severe hydronephrosis with inflammatory changes on both sides and an irregular tumorous mass in the urinary bladder and the right ureter. Following the puncture of the right pelvis of the ureter, the aspiration of 10 mL purulent fluid suggests pyelonephritis. Considering the clinical information (severely reduced glomerular filtration rate, elevated creatinine, metabolic acidosis) without the presence of clinically noticeable symptoms of an airway infection, the cause of death was determined as a post-renal kidney failure due to a formerly unknown carcinoma of the urinary bladder (histologically confirmed) and right ureter (as suspected due to the ultrasound findings). However, it should be noted that the brain was not examined and any cerebral pathologies may not have been recorded or evaluated.

## 4. Discussion

This case report demonstrates the potential to determine underlying diseases causing death by applying post mortem minimally invasive procedures, especially when there is more than one pathological condition allowing for a variety of further investigation methods such as histology, cytology, microbiology, etc. The procedure can be performed in a similar time period as a conventional autopsy or, when done in specialized centers, even faster. Because it does not involve opening of the body cavities and thus inevitable mutilation of the body, MITS is expected to be more accepted by relatives and hence can contribute to reversing the declining autopsy rate worldwide. In MITS, incisions are mostly limited to the almost inconspicuous insertion of a biopsy needle, which contributes to the ‘untouched’ integrity of the deceased [24]. Unlike conventional autopsies, which have a limited possibility for a second examination, MITS can be repeated several times when performed within a certain time interval before decomposition proceeds [1]. MITS is a combined form of post mortem imaging and subsequent needle biopsy tissue sampling. By means of different minimally invasive measures, conventional autopsy may be replaced in various scenarios, and not only due to the destructive character of the gold standard. This idea was previously summarized as Virtopsy^®^ [25]. However, a post mortem ultrasound with tissue biopsies is not yet an introduced method within the original trademark.

Respiratory tract infections contribute to the global burden of infectious diseases and the world has been holding its breath since the COVID-19 pandemic started in 2020 [4]. With this new situation, post mortem research came into focus and was able to address emerging issues such as organ tropism, risk factors for fatal outcome, causes of death, and the infectivity of the corpses and their surfaces [10,11,13].

In the presented case, a combined post mortem ultrasound examination of the major organs followed by tissue sampling using ultrasound guidance with histological follow-up was performed and was able to (i) explain the patient’s clinical presentation and death due to a urogenital malignancy with subsequent post renal failure, (ii) obtain a diagnosis not known during life-time but which well-explained the reduced general state of this patient and (iii) rule out a significant burden of SARS-CoV-2 positivity on this special death circumstance. We were able to detect multiple pathologies including bilateral hydronephrosis resulting from an outlet obstruction by a tumorous mass in the urinary bladder. Moreover, sono-morphological and histological signs of the clinical asymptomatic (at the time of hospital admission) SARS-CoV-2 infection were congruent. The visualization of the severe hydronephrosis during a post mortem examination especially highlights the value of an ultrasound examination in post mortem diagnosis. It can be used as an easy-to-use, cheap, and transportable imaging tool depicting pathological conditions even after death.

Although not limited to, MITS is preferred in investigations of infectious disease cases, and of course is not limited to SARS-CoV-2-associated deaths only but is also applicable to all other emerging diseases now and in the future. Due to lesser contact with contagious material compared to a conventional autopsy, the risk of infections for medical staff is reduced. We were further able to establish a MITS protocol using post mortem CT instead of ultrasound and were successful in the first attempts of robotic tissue samplings to minimize the infectious hazards once more [26].

Despite the many advantages of MITS, some limitations must be considered. Firstly, the post mortem ultrasound examination is temperature dependent. The body should be close to ambient temperature as the cooled fatty tissue seems to restrict the quality of the ultrasound images [27]. Moreover, training for the application and interpretation of ultrasound images is necessary. Secondly, once putrefaction sets in and gas starts accumulating within the body cavities visibility is blurred, and ultrasound-guided tissue sampling as well as sono-morphological evaluation becomes almost impossible [28]. Ultrasound evaluation post mortem is, therefore, limited to the early post mortem period such as in hospital cases. Depending on which ultrasound probe and device is applied, obesity can be a limiting factor [29]. Moreover, it is important to state that only a small area of an entire organ is punctured and thus examined histologically, which can lead to false-negative results if the main pathology has not been sampled itself. To sample brain tissue an additional access to the skull is necessary. Furthermore, the detailed description of organ injuries after trauma is limited compared to other imaging techniques like computed tomography.

## 5. Conclusions

Although conventional autopsy is currently the gold standard of post mortem diagnosis, MITS has been shown to be a very useful tool in post mortem examination practice both in low- and middle-income countries and in times dealing with infectious diseases such as the global COVID-19 pandemic when the safety of medical staff and the environment is necessary or conventional autopsies are negated for several reasons [2,17,24,28,29,30,31]. The future implementation of ultrasound and MITS in the daily routine of forensic and pathological institutes is preferable and requires adequate training and experience.

## Figures and Tables

**Figure 1 diagnostics-13-02643-f001:**
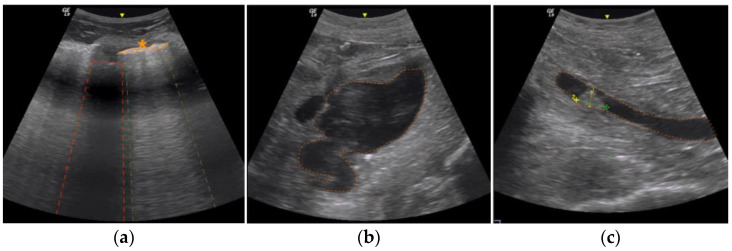
Post mortem ultrasound findings of the lung (**a**), right kidney (**b**) and the right ureter (**c**). (**a**) Right pleural cavity, posteriorly: Pleural lines were thickened and fragmented on both sides (orange asterisk and circle). In addition, B-lines are visible (surrounded by green dotted lines; acoustic shadow of the ribs surrounded by red dotted lines). (**b**) Right kidney: Markedly dilated left kidney pelvis (orange dotted line) resulting from lower urinary tract outlet obstruction. A tumor-like lesion is not visible here. (**c**) Right ureter: In the right ureter (marked with orange dotted lines), which was also dilated, a tumorous obstruction (yellow and green crossed lines) was detected.

**Figure 2 diagnostics-13-02643-f002:**
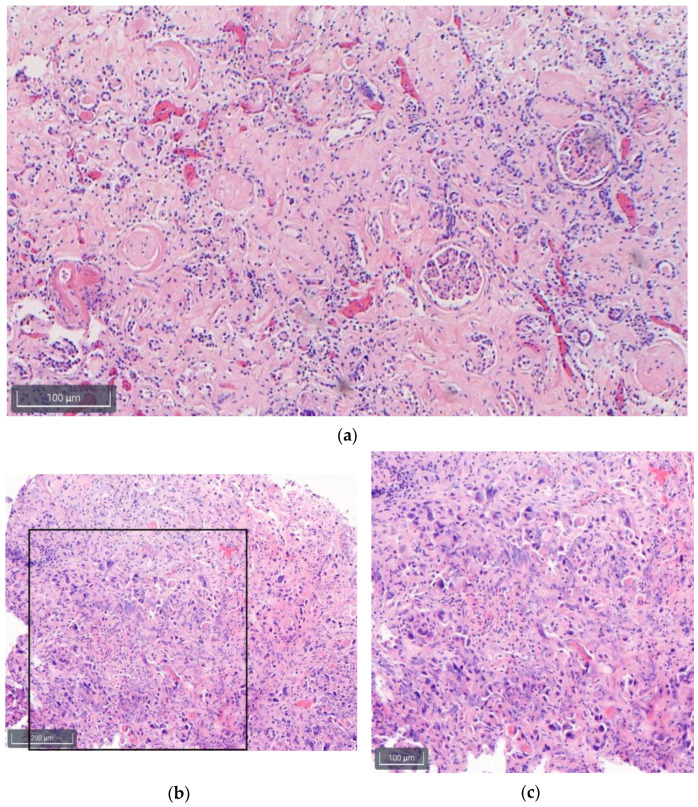
Histological findings of the kidneys (**a**) and the urinary bladder (**b**,**c**) in HE, scale bars bottom left are in different magnifications ((**a**,**c**): 100×; (**b**): 40×). (**a**) Kidneys with chronic nephritis and severe glomerulosclerosis. (**b**) Urinary bladder with a malignant lesion that is composed of admixed high-grade carcinomatous and sarcomatous elements; (**c**) shows histological findings indicated by the box in 2b in more detail. Please note the mixture of both carcinomatous and sarcomatous elements.

## Data Availability

The datasets generated and analyzed during the current study are available from the corresponding author on reasonable request.

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
