# Peer review of "Minimally Invasive Tissue Sampling via Post Mortem Ultrasound: A Feasible Tool (Not Only) in Infectious Diseases—A Case Report"

_diagnostics, 2023, doi:10.3390/diagnostics13162643_

Round 1

Reviewer 1 Report

I believe this was an attempt to underscore the potential to determine diseases causing death by applying minimally invasive procedures especially when there is more than one coexisting pathological condition. I, for one, am a great supporter of the implementation of clinical attainments in postmortem healthcare and minimally invasive tissue sampling is certainly one such example. However, the characteristics of the patient presented here may not be the most appropriate for a “case report”.

 Even if this is passed over (the patient’s age), there is still a poor clinical presentation as a major misgiving. Take, for instance, ultrasound findings listed in sec. 3.1.; while it is possible to understand why were all the US findings given together, as a heterogenous, chaotic pile; it is not evident whether that was part of a clinical assessment or it was simply a modality of PM imaging. I wonder whether the authors are aware of the possibility of using US as the modality of PM imaging.

  It doesn't even have to be talked about the poor clinical correlation of these findings (or “interpretation” if this was part of the clinical work?). For example, we are informed of the “…right kidney showed an intensively dilated pelvis and the left a massive dilatation of the pelvis…” (lns. 133/134)) – but, authors should share their viewpoints on the clinical relevance of these findings. How do cortical thickening and pyelocaliceal dilatation fit into the HTA history?

Honestly, I think that caption to the “figure 1” needs serious reorganization and as for the “figure 2” – image labelled as (b) is totally irrelevant. What is relevant – magnification. The staining by the “Haematoxylin and Eosin” is abbreviated, by most conventions as HE.

All subsections of the “Materials and Methods” lack some basic technical data. I.e., US instrument/ probe details; staining; premortem/ postmortem…

Minor editing is needed.

Reviewer 2 Report

In "Minimally invasive tissue sampling: a feasible tool (not only) 2 in infectious diseases – A case report", the Authors report a case of minimally invasive tissue sampling applied to a woman who died during infection by Sars-SoV-2. The Authors show how this tool may be useful to determine cause of death. The paper is well written and the figures are high-quality. 

Reviewer 3 Report

The article deals with a current and interesting topic, especially after the recent pandemic, during which several issues have been established in the necessary safety protocols for autopsies on COVID-19 subjects.

However, I believe that it would be appropriate to enrich the introduction and discussion by addressing the pandemic context, the issues that have emerged, and the various proposed solutions for the better development of the article.

I suggest talking about the Coivd 19 virus and autopsies and presenting different kind o method that was used as an alternative to conventional autopsies.

Here are some references that could enrich the paper.

Persistence of SARS-CoV-2 viral RNA in nasopharyngeal swabs after death: An observational study

Servadei, F., Mauriello, S., Scimeca, M., ...Schillaci, O., Mauriello, A. Microorganisms, 2021, 9(4), 800  

Post-mortem rt-pcr assay for sars-cov-2 rna in covid-19 patients’ corneal epithelium, conjunctival and nasopharyngeal swabs

Aiello, F., Ciotti, M., Afflitto, G.G., ...Marsella, L.T., Mancino, R. Journal of Clinical Medicine, 2021, 10(18), 4256

Round 2

Reviewer 1 Report

Regardless of how much I do support this paper'smain message, I do not feel that the authors used the best case to report (a 97 year old patient). I suggest to repeat the whole case with a younger participant.
